# Hot Deformation Behavior of an As-Extruded Mg-2.5Zn-4Y Alloy Containing LPSO Phases

Guoxin Wang [1,2], Pingli Mao [1,2,*], Zhi Wang [1,2], Le Zhou [1,2], Feng Wang [1,2] and Zheng Liu [1,2]

1 School of Materials Science and Engineering, Shenyang University of Technology, Shenyang 110870, China; wangguoxin1984@hotmail.com (G.W.); wangzhi8303@163.com (Z.W.); zhoule@sut.edu.cn (L.Z.); wf9709@126.com (F.W.); zliu4321@126.com (Z.L.)

2 Key Laboratory of Magnesium Alloys and the Processing Technology of Liaoning Province, Shenyang 110870, China

* Correspondence: maopl@sut.edu.cn

**Abstract:** The hot deformation and dynamic recrystallization (DRX) characteristics of an as-extruded Mg-2.5Zn-4Y alloy containing long-period stacking ordered (LPSO) phases were investigated using a Gleeble 3500 thermal simulator at temperatures (300–400 °C) and strain rates (0.001–1 s−1). The results revealed that low flow stress corresponded to a high temperature and a low strain rate. An increase in the temperature of deformation caused an increase in the amount of dynamic recrystallization. Additionally, as the strain rate decreased at a given deformation temperature, dislocations were less likely to cause pile-up and dynamic recrystallization was more appropriate, resulting in a lower stress value. Kink deformation was clearly minimized as the number of dynamic recrystallizations increased. The test alloy's activation energy value was determined as 212.144 kJ/mol.

**Keywords:** magnesium alloy; hot deformation; long-period stacking ordered (LPSO); dynamic recrystallization (DRX)





## 1. Introduction

Magnesium alloys are a class of nonferrous metals that are widely used in the domains of national defense, military industries, aircraft, automotive, and electronics where weight is a consideration because of their high strength properties, low density, and superior thermal conductivity [1]. Mg-rare earth alloys are a type of alloy and have gained increasing global attention due to their excellent mechanical properties [2,3]. Recently, magnesium alloys containing the LPSO phases have also attracted attention, especially because they have excellent mechanical properties. LPSO phases have the same (0001) basal plane as Mg, but have an 18-fold longer stacking periodicity along the c-axis [4]. Luo et al. reported that the observed 18R structure is similar to the X-$Mg_{12}$YZn phase [5].

According to Hagiwara et al. LPSO phases of the Mg97Zn1Y2 alloy can promote grain refinement, hence increasing the alloy's strength. These phases can also be used as hardening phases [6]. Kim et al. demonstrated that $Mg_{97}Zn_1Y_1RE_1$ alloys containing LPSO phases have superior mechanical characteristics than alloys devoid of LPSO phases [7]. Tong et al. have shown that mechanical properties of the Mg-Zn-Y magnesium alloy were enhanced due to the strengthening effect of LPSO phases [8]. Peng, C. found that LPSO phases can boost elongation by 42% while maintaining excellent tensile strength [9]. According to Hagihara et al., the presence of LPSO phases promotes extensive grain refining of the Mg matrix at the time of the extrusion forming process, resulting in a considerable increase in yield stress via the Hall–Petch relationship. Additionally, the hardening phase is comprised of LPSO phases running parallel to the extrusion direction [10]. Hagihara et al. found that kinking is a significant deformation mechanism capable of accommodating high levels of stress [11]. It was observed that kinking of LPSO phases results in a significant amount of plastic deformation, which contributes to the materials' ductility improvement.

Garces et al. demonstrated that kinking of LPSO phases increases plastic deformation [12]. Zhang et al. discovered that kinking of LPSO phases and DRX can coordinate the plastic deformation, and kinking of LPSO phases can delay the DRX at 350–450 °C [13]. Zhang et al. found that continuous DRX occurred and observed that the size of the DRX grew as the deformation temperature increased [14]. According to Hao et al., the formation of 14H LPSO phases before extrusion reduced the coarsening of DRX grains in the extruded Mg94Zn2.5Y2.5Mn1 alloy [15].

It is well established that the virgin microstructure of the components has a significant influence on the mechanical characteristics of the resulting Mg alloy during plastic deformation [16]. Nevertheless, the vast majority of previous research has revealed information about the hot working of the primary cast or homogenized Mg-Zn-Y alloys [17–20], few systematic studies have been carried out on the hot working of an as-extruded Mg-Zn-Y alloy. Hence, this study aimed to investigate the stress–strain curves, microstructural evolution, and constitutive equation of an as-extruded Mg-2.5Zn-4Y alloy in order to determine its adaptability to hot work. This study may provide significant experimental data for more in-depth research on large-scale plastic processing of an as-extruded Mg-2.5Zn-4Y including LPSO phases to guide industrial production.

## 2. Experimental Procedures

For this experiment, high-quality Mg (99.95%, wt.%), Zn (99.99%, wt.%), and Mg-Y (20%, wt.%) were used as raw materials. A Mg-2.5Zn-4Y magnesium alloy ingot was cast in a stainless-steel cylindrical steel mold with an inner diameter of 130 mm using an MRL-8 type crucible resistance furnace under the protection of mixed gas (2% $SF_6$ + 98% $N_2$) at melting and casting temperatures of 750 °C and 720 °C, respectively. The alloy ingot was then homogenized for 10 h at 450 °C prior to extrusion. The extrusion process was carried out at 400 °C with an equipment capacity of 1200 T and an extrusion ratio of 6.25. The actual compositions of Mg–2.5Zn–4Y alloys are listed in Table 1.

**Table 1.** The actual compositions of an as-extruded Mg-2.5Zn-4Y (wt%).

| Alloy Composition | Mg | Y | Zn |
|---|---|---|---|
| Mg–2.5Zn–4Y | Bal. | 3.89 | 2.41 |

A wire cutting machine was used to fabricate cylindrical samples with a diameter of φ 8 × 12 mm in the extrusion direction (ED). Isothermal hot-compression studies were carried out on a thermomechanical simulator (Gleeble-3500) at various temperatures (300 °C, 350 °C, and 400 °C) and strain rates (1 $s^{-1}$, 0.1 $s^{-1}$, 0.01 $s^{-1}$, and 0.001 $s^{-1}$), since magnesium alloys are difficult to deform at temperatures below 300 °C. A solid lubricant comprising graphite was put between the samples and indenters to reduce friction. To ensure that the temperature was uniform and stable, the specimens were kept isothermal for 5 min before the hot-compression test. Following that, the samples' deformation microstructures were preserved by rapid quenching in water.

After compression, the microstructure surface sectioned in the center of the specimen was parallel to the compression direction. An optical microscope (OM, Zeiss Axio Observer Z1), a scanning electron microscope (SEM, Zeiss MERLIN Compact), and electron backscatter diffraction (EBSD, EDAX-TSL) were used to examine the microstructure and phase morphology of the Mg-2.5Zn-4Y alloy. These samples were ground, polished, and then etched in a corrosion solution (4.5 g picric acid, 70 mL ethanol, 10 mL 99% acetic acid, and 10 mL of deionized water) before OM and SEM examination. Following that, EBSD specimens were electrolyzed for 60 s at 8 V and −30 °C in an electrolytic solution (10% perchloric acid and 90% ethanol).

## 3. Results and Discussion

### 3.1. The Microstructure of an As-Extruded Mg-2.5Zn-4Y Alloy

The microstructure of an as-extruded Mg-2.5Zn-4Y alloy was depicted in Figure 1. As shown in Figure 1a,b, the mean grain size of the as-extruded Mg-2.5Zn-4Y alloy was around 1.7 μm after extrusion. The second phase is also depicted in Figure 1a with a discontinued net structure that was largely distributed at grain boundaries. The second-phase atomic ratio of Y/Zn ($Mg_{12}ZnY$) was approximately 4/3, as presented in Figure 1c. The theoretical Y/Zn atomic ratio for the $Mg_{12}ZnY$ phase produced in the Mg-Zn-Y alloy was 4/3, and the experimental values were very near to this ratio. In conclusion, the gray second phase was the $Mg_{12}ZnY$ phase, indicating that the second phase is the LPSO phase, as indicated by the arrow in Figure 1a,c.

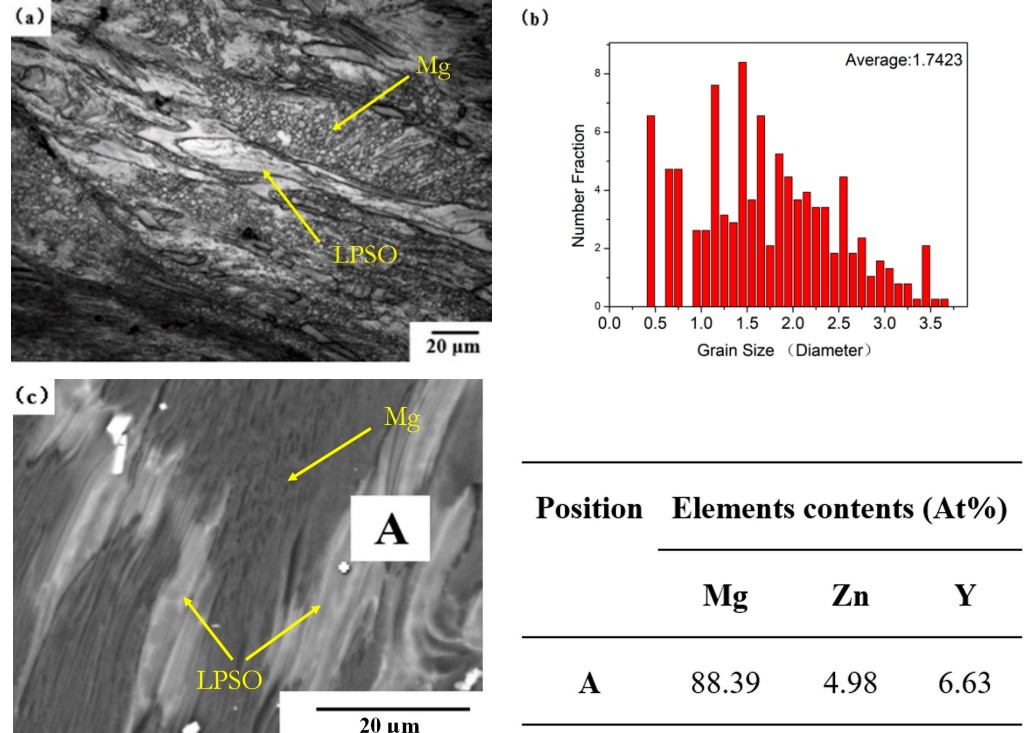

| Position | Elements contents (At%) | | |
|---|---|---|---|
| | **Mg** | **Zn** | **Y** |
| **A** | 88.39 | 4.98 | 6.63 |

**Figure 1.** Microstructure of an as-extruded Mg–2.5Zn–4Y alloy containing LPSO phases: (**a**) OM image; (**b**) average grain size map; (**c**) SEM image and EDS point analysis results.

### 3.2. True-Stress–True-Strain Curves of an As-Extruded Mg-2.5Zn-4Y Alloy

The true-stress–true-strain curves for the as-extruded Mg-2.5Zn-4Y magnesium alloy at several strain rates and various temperatures are depicted in Figure 2. It is worth noting that both the temperature of the deformation and the strain rate has a significant effect on the flow behavior. Figure 2 depicts the variability in peak flow stress during various deformation conditions. As illustrated, the flow stress reduces as the strain rate decreases for a given temperature of deformation, whereas it increases as the deformation temperature decreases for a given strain rate. Furthermore, as the strain rate decreases at a given deformation temperature, dislocations are less likely to cause pile-up, and dynamic recrystallization becomes more appropriate, resulting in a lower stress value. The overall characteristics of true-stress–true-strain curves were comparable under all deformation conditions (Figure 2a,b). As the strain increased, the stress also increased abruptly, and the stress reached its maximum as the strain rose further. Finally, as the strain increased, the flow stress reduced and the system achieved a steady state. Figure 2c revealed that stress increased with strain and that stress gradually increased to its maximum with further strain increase, although the curve had no clear steady state. Figure 2d demonstrated

that the stress rapidly increased with increasing strain and had already achieved a steady state when the strain was increased further to 0.05, indicating that there was no obvious peak stress.

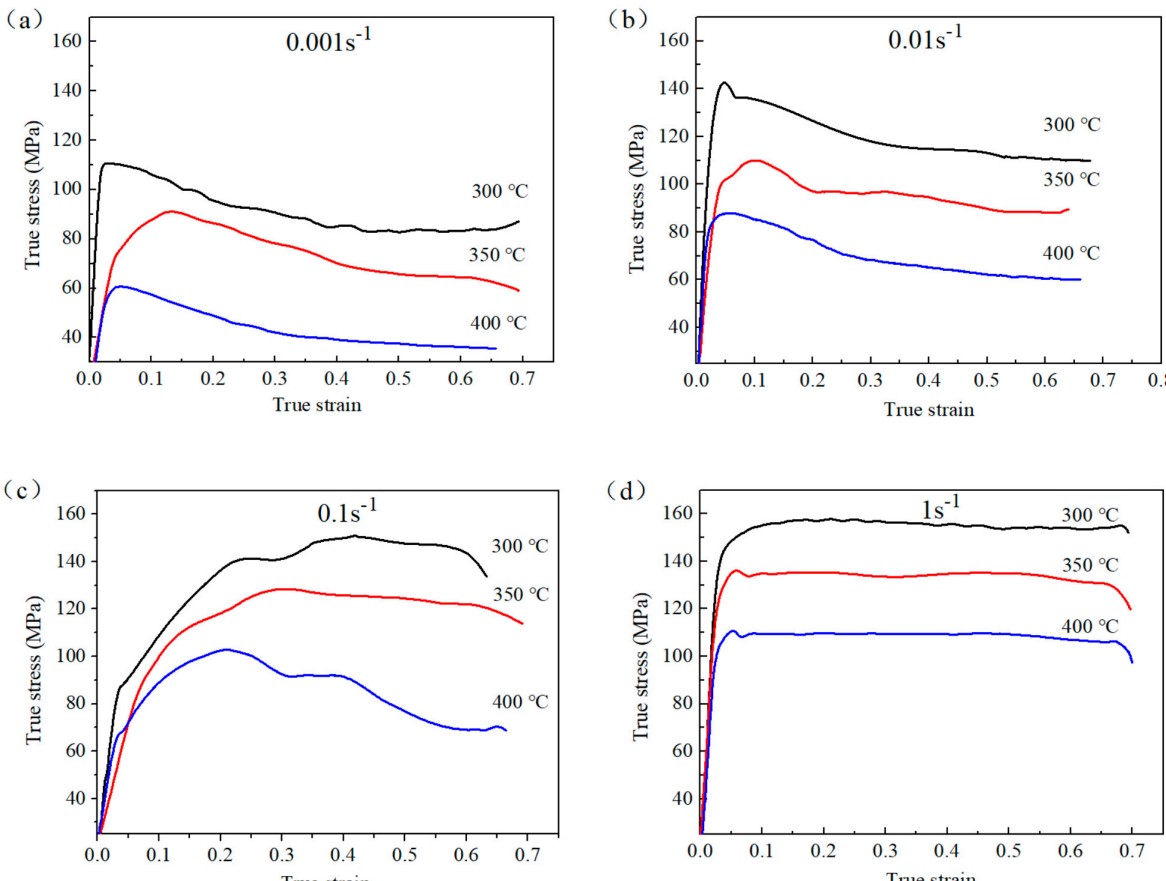

**Figure 2.** True-stress–true-strain curves at several strain rates and temperatures (**a**) 0.001 s$^{-1}$, 300–400 °C; (**b**) 0.01 s$^{-1}$, 300–400 °C; (**c**) 0.1 s$^{-1}$, 300–400 °C; and (**d**) 1 s$^{-1}$, 300–400 °C.

Under the combined action of dynamic recrystallization and strain hardening, the softening effect of magnesium alloy exceeds the work-hardening effect [21,22]. Hence, the flow stress of compression deformation lowered as the compression strain increased. The flow stress drops slightly with increasing strain (Figure 2a–c) or remains essentially constant (Figure 2d). At this point, the softening and hardening effects of magnesium alloy have reached a state of dynamic equilibrium at this point.

### 3.3. DRX Behaviour of an As-Extruded Mg-2.5Zn-4Y Alloy

Figure 3 depicts as-extruded Mg-2.5Zn-4Y alloy optical micrographs at various temperatures and strain rates (0.001 s$^{-1}$). As illustrated in Figure 1a, the grain size (1.7 μm) was extremely small after extrusion. The tendency for fine-grain development becomes quite obvious when held at a high temperature for five minutes (as illustrated in Figure 3a). LPSO phases can impact the dynamic recrystallization formation process due to their extraordinary thermal stability. Figure 3b is an expanded view of the white rectangular area in Figure 3a. As illustrated in Figure 3b, a minor proportion of recrystallized grains can be observed along the grain boundary of elongated grains, as indicated by the white rectangle in Figure 3b, resulting in the necklace structure characteristic of continuous dynamic recrystallization (CDRX) [23]. Simultaneously, as indicated by the white arrow in Figure 3b, a few DRX grains developed at the interface of the phase between the Mg matrix and bulk LPSO. This might be because of LPSO phases promoting the DRX through a particle-stimulated nucleation (PSN) mechanism [24].

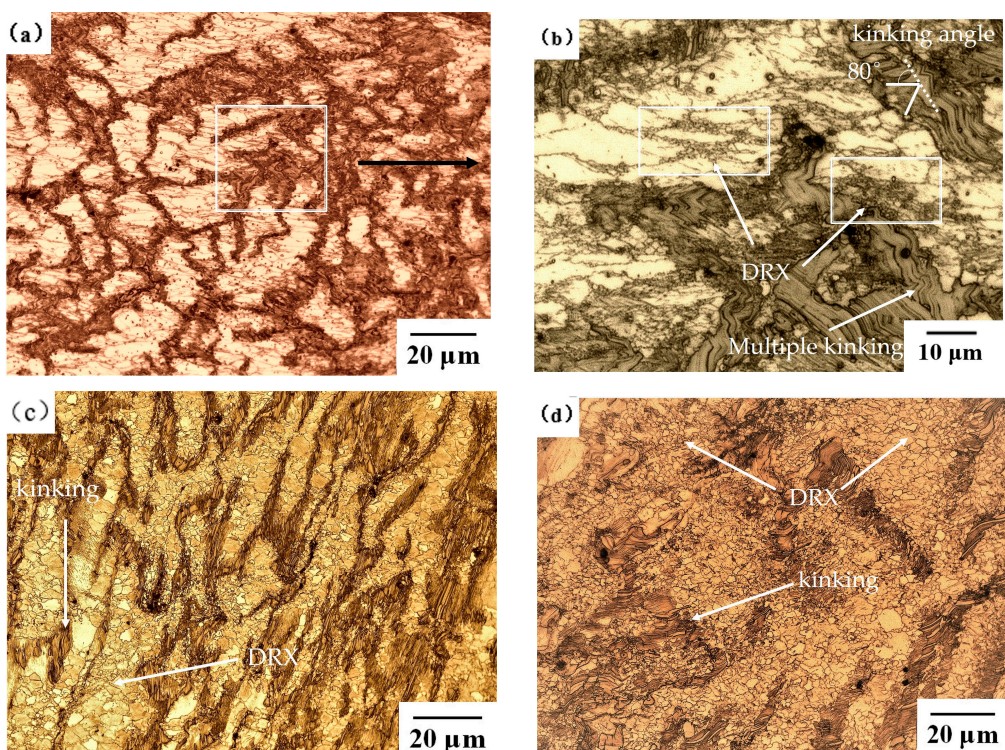

**Figure 3.** OM images exhibiting DRX and kinking within deformed alloys at a strain rate of 0.001 s$^{-1}$ at several temperatures: (**a**,**b**) 300 °C; (**c**) 350 °C; and (**d**) 400 °C.

All delocalizations were easily stretched and entangled because of the low stacking fault energy of magnesium alloys, generating a dense network of dislocations on the slip surface. Due to the difficulty of clustering a partial dislocation by sliding and climbing [25,26], dynamic recovery was delayed in the presence of recrystallization. As a result, the hardening process can be considered predominant. Additionally, multiple kinking and a large kinking-angle characteristic of LPSO phases can be shown by the white arrow in Figure 3b. Simultaneous dislocation pair slip occurs on the basal planes of LPSO phases, resulting in phase kinking. The kinking of LPSO phases is suggested to improve the alloy's strength. Because of the poor symmetry and the less active slip mechanisms, non-uniform deformation of magnesium alloy will occur [27]. The LPSO kink band can coordinate deformation under certain conditions. Due to the low formation temperature of 300 °C, DRX's nuclear power was repressed. Despite the low DRX content, this sample demonstrated substantial flow stress, reaching a maximum of 110.69 MPa. Cross-slip of screw dislocations and climbing of non-basal planes were more difficult to produce at low temperatures than at high temperatures. Low temperatures are unfavorable for recrystallization nucleation because the driving force produced by low dislocation density cannot achieve the critical flow stress required for dynamic recrystallization. The number of independent slip systems increases as the temperature rises, atoms and vacancies disperse due to thermal activation, and the grain size of recrystallization increases significantly due to grain boundary migration, which favors recrystallization nucleation and growth [28,29]. The DRX volume fraction steadily increased to 300 °C and 400 °C in Figure 3. This was also the explanation for the decrease in flow stress at 300 °C, 350 °C, and 400 °C, as schematically shown in Figure 2a.

In comparison to 300 and 350 °C, the recrystallized grain size and fraction increased significantly at 400 °C (as illustrated in Figure 3d). Multiple kinking and a large kinking angle, on the other hand, were not apparent (as shown by the blue rectangular area). With an increase in the number of dynamic recrystallizations, kink deformation decreased dramatically. Above all, the kinking of LPSO phases and dynamic recrystallization contribute significantly to the alloy's softening under hot compression.

Figure 4 depicts the alloy micrographs at various strain rates at 400 °C. The dynamic recrystallization degree was very modest or did not occur at any temperatures under this strain rate, as depicted in Figure 4a. This was the reason why the curve in Figure 2d lacked peak stress. Figure 4b is an expanded view of the rectangular black area in Figure 4a. It is clear from Figure 4b that a large number of twins coexisted with kinked LPSO coordinate deformation, and dynamic recrystallization was rare under these conditions. Figure 4c shows that a few recrystallized grains were formed on the large grain boundaries, and the alloy structure formed a necklace structure surrounded by small recrystallized grains. At this time, the twins were eaten away by the recrystallized grains and could not be distinguished.

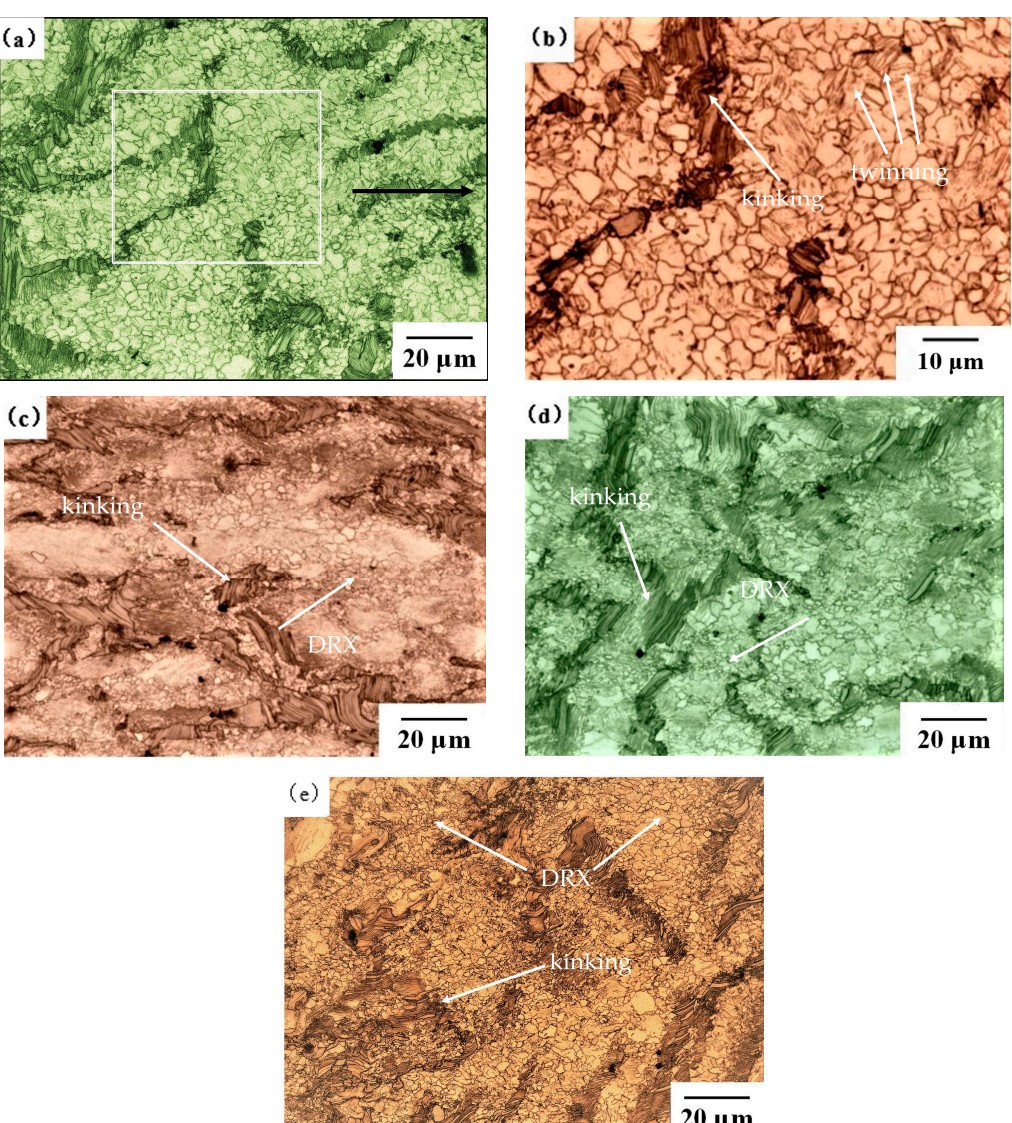

**Figure 4.** OM images of the DRX and kinking inside the deformed alloys at 400 °C and several strain rates: (**a**,**b**)1 s$^{-1}$; (**c**) 0.1 s$^{-1}$; (**d**) 0.01 s$^{-1}$; and (**e**) 0.001 s$^{-1}$.

As illustrated in Figure 4d, the alloy forms a high number of fine recrystallized grains. Reduced strain rate to 0.001 s$^{-1}$ resulted in the formation of more fine and uniform recrystallization grains (as illustrated in Figure 5e).

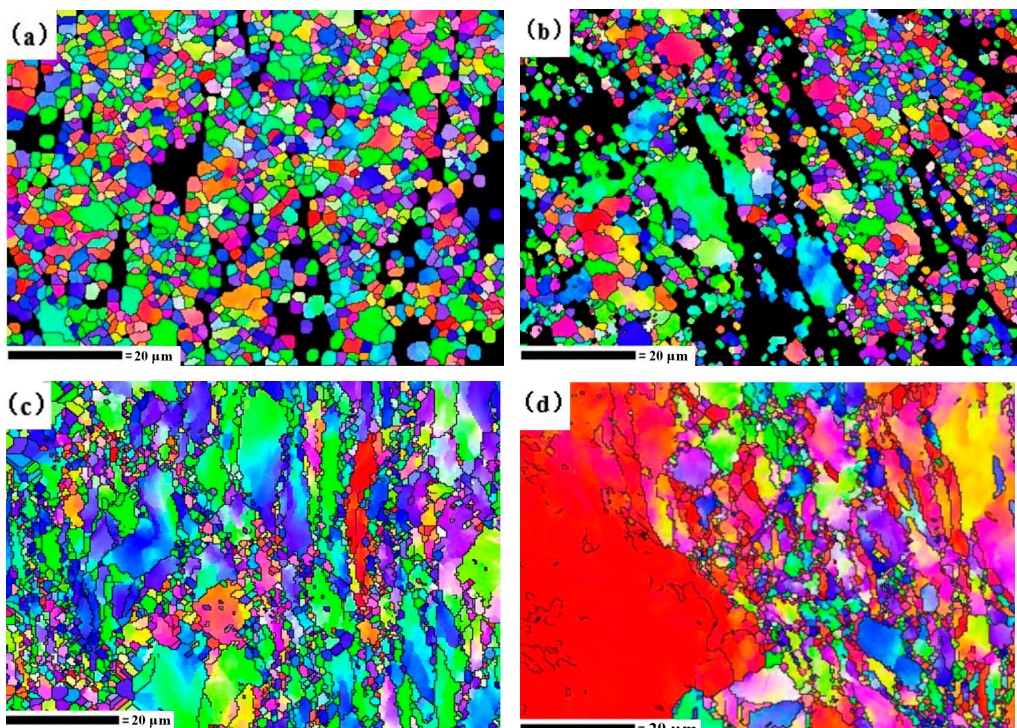

**Figure 5.** The IPF maps (**a–d**) of compressed samples under different conditions: (**a**) 400 °C/0.001 s$^{-1}$; (**b**) 350 °C/0.001 s$^{-1}$; (**c**) 300°C/0.001 s$^{-1}$; and (**d**) 300 °C/0.01 s$^{-1}$.

Figure 4a–e shows that when the strain rate is reduced at a given deformation temperature (400 °C), dislocations are less likely to cause pile-up and dynamic recrystallization is more appropriate. The degree of kinking of LPSO phases at different strain rates is similar, implying that the degree of coordination of magnesium alloy deformation for LPSO phases is nearly the same.

Figure 5 shows typical IPF maps for samples compressed at temperatures (400 °C, 350 °C, and 300 °C) with strain rates (0.01 s$^{-1}$ and 0.001 s$^{-1}$). As can be seen, the DRX improved with the increase in temperature and a reduction in strain rate. Except for the entire DRX at 400 °C/0.001 s$^{-1}$, other specimens only had partial DRX. The black spots indicate LPSO phases with a block-shaped structure, which cannot be examined by EBSD. The fundamental reason for this is that the Kikuchi lines of the Mg matrix and the LPSO phases overlap, making indexing LPSO difficult. The DRX grains were scattered along the original grain boundaries (as seen in Figure 5c,d). As the primary method of softening, the presence of DRX decreased distortion within coarse unDRXed grains. The DRX grains have grown sufficiently to become uniform, as indicated in Figure 5a. This is because of the suitably high temperature of deformation and the sufficiently lower strain rate. As a result, the DRX nucleus has sufficient time to develop. It mostly resulted in the dynamic recrystallized grains formation. When the strain rate was raised and the deformation temperature was reduced, the time for DRX grain development and nucleation decreased, resulting in deformed grains that were not completely recrystallized, as seen in Figure 5c,d.

### 3.4. Constitutive Equation

To explore as-extruded Mg-2.5Zn-4Y alloy thermal deformation behavior, the constitutive equation was used to establish a connection between deformation temperature, flow stress, and strain rate [30–32]. The hyperbolic sine–type equation (Equation (1)) is widely accepted for accurately describing the flow behavior of magnesium alloys amongst parameters of deformation for a broad range of stresses [32]:

$$\dot{\varepsilon} = A[\sinh(\alpha\sigma)]^n \exp(-Q/RT) \tag{1}$$

where σ, Q, T, and R represent the flow stress, the deformation activation energy in kJ/mol, the absolute temperature in K, and the universal gas constant (8.314 J/(mol·K)). A, n, and α are flow stress and deformation-temperature-independent constants of the material. $\dot{\varepsilon}$ denotes the strain rate and Z denotes the parameter of Zener–Hollomon. When there is low flow stress (ασ < 0.8), exponential law can simplify Equation (1) to [33]:

$$\dot{\varepsilon} = A_1 \sigma^{n_1} \exp(-Q/RT), \ \alpha\sigma < 0.8 \tag{2}$$

When there is low flow stress (ασ > 1.2), exponential law can simplify Equation (1) to [33]:

$$\dot{\varepsilon} = A_2 \exp(\beta\sigma) \exp(-Q/RT), \ \alpha\sigma > 1.2 \tag{3}$$

For equation simplification, the natural logarithm was taken on both sides of two Equations (2) and (3). This could be utilized to determine the n and β values.

$$\ln \dot{\varepsilon} = B_1 + n_1 \ln \sigma \tag{4}$$

$$\ln \dot{\varepsilon} = B_2 + \beta\sigma \tag{5}$$

According to Equation (4), the value of $n_1$ can be calculated from the slope of the line in $\ln \dot{\varepsilon} - \ln \sigma$ plots and the value of β can be calculated from the lines' average slope in $\ln \dot{\varepsilon} - \sigma$ plots based on Equation (5). The linear relationships of $\ln \dot{\varepsilon} - \ln \sigma$ and $\ln \dot{\varepsilon} - \sigma$ were fitted at various temperatures, as depicted in Figure 6. Furthermore, the fitted lines have correlation coefficients of at least 0.92 and 0.93.

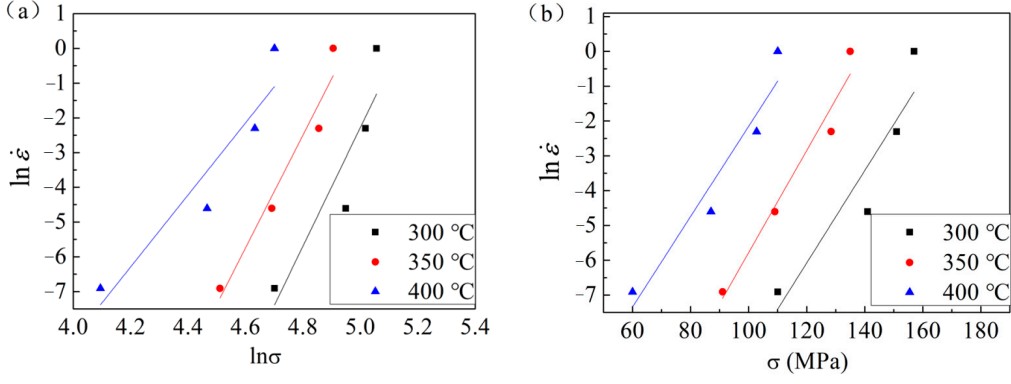

**Figure 6.** The fitting of linear relationships: (**a**) $\ln \dot{\varepsilon} - \ln \sigma$; (**b**) $\ln \dot{\varepsilon} - \sigma$ .

Thus, the $n_1$ and β average values were calculated as 14.567 and 0.136 MPa$^{-1}$, respectively, for the peak data. In this case, the mean value of α was calculated as $9.336 \times 10^{-3}$ MPa$^{-1}$ based on $\alpha = \beta/n_1$.

Assuming that the activation energy of thermal deformation has no relationship with T in a specific temperature range, both sides logarithm of Equation (1) may be obtained as follows:

$$\ln \dot{\varepsilon} = \ln A - Q/RT + n \ln[\sinh(\alpha\sigma)] \tag{6}$$

When the set temperature was fixed, the partial differential of Equation (6) can be obtained, and the calculation formula of activation energy of deformation Q is as follows:

$$Q = R \left\{ \frac{\partial \ln \dot{\varepsilon}}{\partial \ln[\sinh(\alpha\sigma)]} \right\}_T \cdot \left\{ \frac{\partial \ln[\sinh(\alpha\sigma)]}{\partial T^{-1}} \right\}_{\dot{\varepsilon}} \tag{7}$$

Take the peak stress and plot $\ln \dot{\varepsilon} - \ln[\sinh(\alpha\sigma)]$ and $\ln[\sinh(\alpha\sigma)] - T^{-1}$ curves as presented in Figure 7a,b, respectively. According to Equation (7), Q can be expressed as follows: Q = R × C × D. Moreover, C = 10.915, and D = 2.338, respectively, are the averages of all slopes in Figure 6a,b. The deformation activation energy Q was 212.144 kJ/mol, based on the average slope of the two curves. According to the researchers, the relationship

between the temperature of deformation and strain rate can be determined using the Z parameter.

$$Z = \dot{\varepsilon}\exp(Q/RT) = A[\sinh(\alpha\sigma)]^n \tag{8}$$

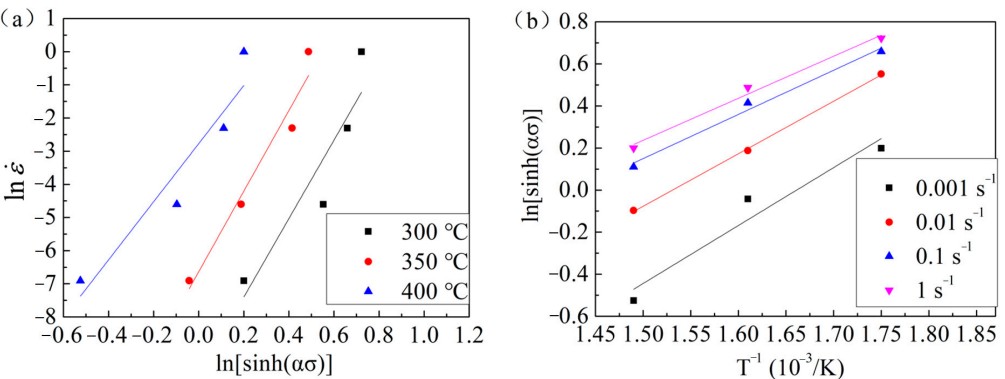

**Figure 7.** Linear relationship fitting: (**a**) $\ln\dot{\varepsilon} - \ln[\sinh(\alpha\sigma)]$ (**b**) $\ln[\sinh(\alpha\sigma)] - 1/T$.

The Z parameter's physical meaning is the factor of strain rate used for temperature compensation. Logarithm was taken on both sides of Equation (9), and thus obtained:

$$\ln Z = n\ln[\sinh(\alpha\sigma)] + \ln A \tag{9}$$

Based on Equation (9), the $\ln Z - \ln[\sinh(\alpha\sigma)]$ linear regression data were obtained at n = 10.602, lnA = 35.145, respectively, as depicted in Figure 8. Thus, the value of A is $1.833 \times 10^{15}$ s$^{-1}$. According to the Arrhenius function definition, σ can be represented as a function of the Z parameter as indicated in Equation (10).

$$\sigma = \frac{1}{\alpha}\ln\left\{\left(\frac{Z}{A}\right)^{\frac{1}{n}} + \left[\left(\frac{Z}{A}\right)^{\frac{2}{n}} + 1\right]^{\frac{1}{2}}\right\} \tag{10}$$

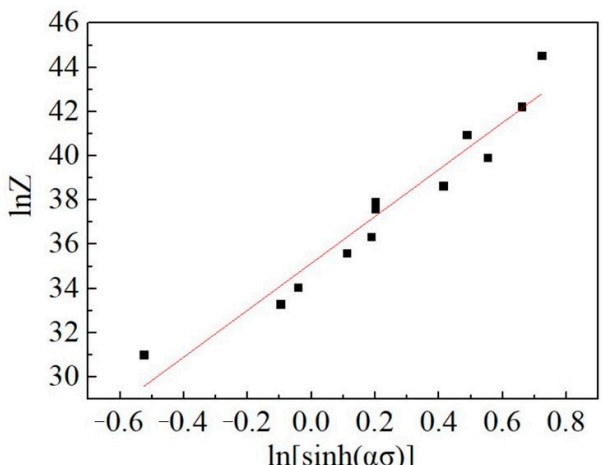

**Figure 8.** Linear relationship between $\ln Z$ and $\ln[\sinh(\alpha\sigma)]$.

Finally, substituting A, Q, R, α, n values into Equation (1), the Arrhenius flow stress constitutive equation of Mg-2.5Zn-4Y alloy was obtained as:

$$\dot{\varepsilon} = 1.833 \times 10^{15} \times \left[\sinh\left(9.336 \times 10^{-3}\sigma\right)\right]^{10.602} \times \exp\left(-\frac{212.144}{8.314T}\right) \tag{11}$$

The flow stress was described using the Z parameter expression as:

$$\sigma = \frac{1}{9.336 \times 10^{-3}} \times \ln\left\{ \left(\frac{Z}{1.833 \times 10^{15}}\right)^{\frac{1}{10.602}} + \left[ \left(\frac{Z}{1.833 \times 10^{15}}\right)^{\frac{2}{10.602}} + 1 \right]^{\frac{1}{2}} \right\} \quad (12)$$

The activation energy Q of an as-extruded Mg-2.5Zn-4Y alloy is 212.144 kJ/mol, which is much larger than that of pure Mg at 135 kJ/mol. Xia et al. [34] reported the hot deformation behavior of as-extruded Mg-Zn-Y-Zr alloy containing W phases, and they reported Q was determined as 137.400 kJ/mol. This indicates that the LPSO phases are extremely stable and can act as effective barriers, resulting in the test alloy's high Q value.

## 4. Conclusions

The DRX and hot deformation behaviors of an as-extruded Mg-2.5Zn-4Y alloy containing LPSO phases were studied. The main conclusions were drawn as follows:

1. As the strain rate increased, the flow stress also increased at the same temperature, and the flow stress reduced with increasing temperature at the same strain rate. Specifically, there was no obvious peak stress at different temperatures at the strain rate of $1 \text{ s}^{-1}$.
2. With an increase in the number of dynamic recrystallizations, kink deformation decreased dramatically. The kinking of LPSO phases and dynamic recrystallization both contribute significantly to the alloy's softening under hot compression.
3. The number of dynamic recrystallizations increased with the increase in temperature or reducing strain rate. The specimens compressed at $400 \text{ }^{\circ}\text{C}/0.001 \text{ s}^{-1}$ exhibited relatively large DRX ratios and the best working performance.
4. The test alloy's Q value was determined as 212.144 kJ/mol. The constitutive equation of an as-extruded Mg-2.5Zn-4Y alloy with LPSO phases deformed at various deformation temperatures is as follows:

$$\dot{\varepsilon} = 1.833 \times 10^{15} \times \left[ \sinh\left(9.336 \times 10^{-3}\sigma\right) \right]^{10.602} \times \exp\left(-\frac{212.144}{8.314\text{T}}\right) \quad (13)$$

**Author Contributions:** Conceptualization, G.W., P.M. and Z.W.; methodology, G.W., P.M., Z.L. and F.W.; investigation, G.W., P.M., Z.L. and F.W.; data curation, G.W.; writing—original draft preparation, G.W.; writing—review and editing, G.W. and P.M.; supervision, G.W., P.M., Z.L. and L.Z. All authors have read and agreed to the published version of the manuscript.

**Funding:** This research was funded by the High level innovation team of the Liaoning Province (XLYC1908006) and Project of Liaoning Education Department (LZGD2020003). And The APC was funded by Liaoning Revitalization Talents Program (No. XLYC1807021 and 1907007).

**Institutional Review Board Statement:** Not applicable.

**Informed Consent Statement:** Not applicable.

**Data Availability Statement:** Data presented in this study may be requested from the corresponding authors.

**Conflicts of Interest:** The authors declare no conflict of interest.

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
