# Peer review of "Hot Deformation Behavior of an As-Extruded Mg-2.5Zn-4Y Alloy Containing LPSO Phases"

_metals, doi:10.3390/met12040674_

Round 1

Reviewer 1 Report

The present paper is having sound scientific contributions and well written. The authors have taken much effort. However, the authors have to address the following points before it is being considered for its publications

  1. Expansion of LPSO phase should be incorporated in the abstract
  2. Numerical outcome of strength at different temperature are to be incorporated in the abstract
  3. Why the lower value of flow stress observed at lower strain rate? Scientific reasons as a single line is to be incorporated in the abstract. For high temperature, the flow stress value is obviously decreased. It is not necessary to add the reason. However, for the lower strain rate, the reason for decreasing of flow stress value should be addressed
  4. Why the authors have taken the temperature 300 to 400 degress only? Why the authors have not taken at room temperature, and lower than 300 degrees. It has to be addressed in the manuscript.
  5. Introduction part is very low. More literature is to be added for addressing the hot deformation behaviour of other Mg based systems
  6. More explanation related to kinking phenomenon is to be addressed. The kinking phenomenon will usually occurs in composites. How, the kinking phenomena occurs in isotropic metals?. Needs more clarification.
  7. Various phases observed in microstructures of Figure 1 is to be marked inside the microstructures (Figure 1 a and c)
  8. In Figure 2, all the plots are to be drawn in a same scale so that the authors/readers can ses the differences with temperature and strain rate.
  9. Similar to comment no 7, various microstructural features are to be incorporated in Figures 3 and 4 also.
  10. Why black spots/black shaded area are observed in EBSD coloured map of Figure 5? Need clarification

Reviewer 2 Report

The paper has a lot of mistakes that are detailed in the attached file.

The most important aspects to revise are:

1) To describe with more details the experimental procedure, what kind of machine was used?

2) A clear description of LPSO phases is needed: crystallography, stoichiometry etc.

3) Equation 7  is not clear

4) What are the correlation factors In Fig. 6?

5)It is not clear how the authors computed the different parameters of the model

6) In 3.2, the first paragraph contains a sentence that it is wrong

7) Minor mistakes are detailed in the text

Kind regards

Reviewer 3 Report

It seems that the novelty of this investigation is poor and cannot satisfy the scientific needs of professionals in this field. In addition, the characterization of the resultant products and the investigated properties seems to be insufficient. Authors are recommended to provide more scientific discussions and more experimental results. Unfortunately, the presented manuscript in this condition can not satisfy the journal standard.

Reviewer 4 Report

Dear Authors,

The article contains the results and research on the hot compression experiment to investigate the hot deformation and dynamic recrystallization (DRX) characteristics of the extruded Mg-2.5Zn-4Y alloy containing LPSO phases at temperatures (300 ° C-400 ° C) and the strain rate ( 0.001 s -1–1 s – 1) using the Gleeble 3500 thermal simulator. The purpose is stated: “Hence, the main purpose of this essay was to acquire variation and the corresponding accommodated role of LPSO morphology and the DRX of an as-extruded Mg-2.5Zn-4Y alloy during the hot compression tests. However, the results are presented in a way that makes their analysis difficult (it requires a lot of effort from him), and their discussion does not fully show the scientific aspect:

  1. Lack of description in the methodology of testing devices (their origin) and parameters of the tests performed. For example, the writing that the alloy was made the traditional way may not be understood by everyone
  2. Figure 3 is illegible and labeled. The reader must guess what the author meant.
  3. There are references to drawings, but no indications of either the phases or the mechanisms described are in the drawings. (figure 1.4)
  4. Lack of answers in the applications, do the obtained results improve the properties of the material, how do they affect it? How does the variability and the corresponding role of the LPSO and DRX morphology of the extruded Mg-2.5Zn-4Y alloy during hot compression testing affect the properties and what does it mean for science?

Reviewer 5 Report

1) An additional explanation of the technique for obtaining ingots is required. Raw materials, production method, melting and casting temperature, and so on.
2) What method was used to determine the composition of the resulting alloy?
3) "Hence, the main purpose of this essay was to acquire variation and the corresponding accommodated role of LPSO morphology and the DRX of an as-extruded Mg-2.5Zn-4Y alloy during the hot compression tests." The purpose of the work is not entirely clear. The authors do not provide research in the process of deformation
4) Correction of the conclusions is required, at the moment they are just a description of the results obtained in the article. It is also not clear why the resulting formula is given.

Round 2

Reviewer 1 Report

The authors have worked based on my previous comments and prepared the revised article. Therefore, I am recommending to accept the article

Author Response

Dear reviewer, thank you very much for accepting our article.

Reviewer 2 Report

Some of my suggestions was not made.

Fig2. vertical axis, replace "Mpa" by "MPa"

Line 163: The expression is "stacking fault energy", nor "fault energy of stacking"

Line 176: 158 MPa does not correspond to the maximum stress in fig 3b.

Line 164: the word "dislocation" must be in plural "dislocations"

Line 242: R units are J/molK, not J/K. It is unacceptable that authors do not change these units. 

Fig. 6b. Units in x-axis must be MPa

Conclusion 2: It is not understandable

Last conclusion: a conclusion should not start with "And"

Reviewer 3 Report

It is recommended to improve the introduction section by reporting the qualified previous studies in this field like:
Hot deformation constitutive model and processing maps of homogenized Ale5Mge3Zne1Cu alloy (https://doi.org/10.1016/j.jmrt.2021.06.069).

In addition, please refine the manuscript one more time to improve the English language and style.

Reviewer 4 Report

Dear Author

Some content cannot be read in a file with all the text. Sections 1 and 4 cannot be read at all (I do not know if they are removed?). Further, there are no described phase mechanisms.

Kind regards,

Reviewer 5 Report

Please remove the formula from abstract

Author Response

According to the reviewer’s suggestion,  we have removed the formula from the abstract.